# Fine-scale collective movements reveal present, past and future dynamics of a multilevel society in Przewalski's horses

Katalin Ozogány [1,2] ✉, Viola Kerekes[3], Attila Fülöp [1,2,4,5,6], Zoltán Barta [1,2,11] & Máté Nagy [7,8,9,10,11] ✉

Studying animal societies needs detailed observation of many individuals, but technological advances offer new opportunities in this field. Here, we present a state-of-the-art drone observation of a multilevel herd of Przewalski's horses, consisting of harems (one-male, multifemale groups). We track, in high spatio-temporal resolution, the movements of 238 individually identified horses on drone videos, and combine movement analyses with demographic data from two decades of population monitoring. Analysis of collective movements reveals how the structure of the herd's social network is related to kinship and familiarity of individuals. The network centrality of harems is related to their age and how long the harem stallions have kept harems previously. Harems of genetically related stallions are closer to each other in the network, and female exchange is more frequent between closer harems. High movement similarity of females from different harems predicts becoming harem mates in the future. Our results show that only a few minutes of fine-scale movement tracking combined with high throughput data driven analysis can reveal the structure of a society, reconstruct past group dynamics and predict future ones.

Understanding social structure and dynamics of animal societies is an important task in which utilising emerging technologies and high-throughput methods could be the key. Multilevel societies are arguably among the most complex forms of social organisation in nature[1,2]. Individuals in these societies aggregate through multiple nested levels[2]: the core units (the lowest social level) are usually breeding units (e.g., one-male, multifemale groups), and higher levels of social organisation are formed by the aggregation of lower-level units. They are best known from primates[3-7], but are also found in cetaceans[8,9], elephants[10], equids[11-14], and birds[15]. Moreover, the vast majority of human social systems show multilevel structure as well[16-20], thus studying multilevel societies can fundamentally contribute to our understanding of the evolution of sociality[2,21-23].

[1]ELKH-DE Behavioural Ecology Research Group, University of Debrecen, Egyetem tér 1, Debrecen 4032, Hungary. [2]Department of Evolutionary Zoology and Human Biology, University of Debrecen, Egyetem tér 1, Debrecen 4032, Hungary. [3]Hortobágy National Park Directorate, Sumen u. 2, Debrecen 4024, Hungary. [4]Evolutionary Ecology Group, Hungarian Department of Biology and Ecology, Babeș-Bolyai University, Str. Clinicilor 5-7, 400006 Cluj-Napoca, Romania. [5]Centre for Systems Biology, Biodiversity and Bioresources (3B), Babeș-Bolyai University, Str. Clinicilor 5-7, 400006 Cluj-Napoca, Romania. [6]STAR-UBB Institute of Advanced Studies in Science and Technology, Babeș-Bolyai University, Str. Mihail Kogălniceanu 1, 400084 Cluj-Napoca, Romania. [7]MTA-ELTE "Lendület" Collective Behaviour Research Group, Hungarian Academy of Sciences, Pázmány P. Stny. 1A, Budapest 1117, Hungary. [8]Department of Biological Physics, Eötvös Loránd University, Pázmány P. Stny. 1A, Budapest 1117, Hungary. [9]MTA-ELTE Statistical and Biological Physics Research Group, Hungarian Academy of Sciences, Pázmány P. Stny. 1A, Budapest 1117, Hungary. [10]Department of Collective Behavior, Max Planck Institute of Animal Behavior, Universitätsstraße 10, 78457 Konstanz, Germany. [11]These authors contributed equally: Zoltán Barta, Máté Nagy. ✉e-mail: katalin.ozogany@gmail.com; nagymate@hal.elte.hu

One such taxon where multilevel social organisation can occur is the Przewalski's horse (*Equus ferus przewalskii*)[14,24], which is the last extant subspecies of wild horses (*Equus ferus*). The mating system of Przewalski's horses is female defence polygyny, where year-round stable harems are the core units of the society, and the harem's single breeding male (the harem stallion) protects the adult females and their juvenile offspring belonging to the harem[11,25–27]. Another type of social groups are the single-sex bachelor groups formed by non-breeding adult males[27,28]. Previous observations show variation in the occurrence of higher-level social units in Przewalski's horses: harems are observed to live isolated in almost exclusive home ranges in Hustai National Park[29], while in other populations they aggregate and form multilevel herds[14,24,30]. In Hortobágy National Park, Hungary, where the largest captive population of Przewalski's horses lives in a 3000-ha fenced but otherwise natural habitat, both cases were observed: in the first years after introduction harems had non-overlapping home ranges, while recently the population forms a massive multilevel herd[14]. However, the detailed structure of this society (e.g., the bonds between harems leading to herd formation) remains unclear.

A commonly used approach in the study of social structures is the analysis of social networks, where the links between individuals are traditionally quantified by numerous direct observations of social interactions over a long period of time[31,32]. However, uncovering the structure of a multilevel societies' social network would require the observation of many, possibly as much as a few hundreds of individuals. Recent advances in bio-logging and remote monitoring enables the collection of large amounts of behavioural data over short periods and may involve the majority or even all individuals in a social group, hence affording the quick and reliable study of social structures[33]. New technologies thus offer the possibility of a more detailed analysis of complex societies than ever before, however, a deeper understanding is required on how these detailed "snapshots" of the system reflect relationships among individuals that have developed over longer timescales[34,35].

In this work, we present the results of drone observations to track movements of Przewalski's horses in Hortobágy National Park ($n = 278$ individuals) and combine the high-resolution movement data of several minutes with long-term demographic data collected over 23 years of continuous population monitoring (Fig. 1). We aimed to study the collective movements in this multilevel herd of Przewalski's horses and explore relationships between society structure and motion patterns. We show that the structure of the society (i.e., associations of individuals to harems) can be determined from movements, which we expected from our previous studies[30]. Moreover, by characterising the society with proximity networks during movements, we uncover novel relationships between individuals and harems based on kinship and familiarity, we reveal that network centrality is related to harem traits, and that the network's structure is related to past and future social dynamics, i.e., member exchanges between harems, which exchanges for the future—as an unexpected finding—we can predict from the movements of individuals.

## Results

### Data acquisition techniques

Our aerial observations consisted of 5-min long video sessions captured on five different days. We recorded 4k videos of the herd's movements with two drones simultaneously to get global motion patterns and enough details for individual recognition (Fig. 1a; see Methods). During these observations the herd followed their natural daily routine, moving undisturbed in the reserve. We tracked each individual's movement on the footage (pixel coordinates) and referenced the locations on the images to the background for earth-fixed metric coordinates and thus reconstructed movement trajectories in high temporal (12.5 position/s) and spatial (+/− 0.2 m) resolution (Fig. 1b, c; see Methods). All horses, except bachelor males, were

individually identified ($n = 238$) in the footage and their identities were matched across recording sessions on different days (Fig. 1c; Supplementary Table 1).

Przewalski's horses living in the Pentezug reserve of the Hortobágy National Park typically formed a single compact herd and moved in a very coordinated way during all recording sessions (Fig. 1a, Supplementary Movies 1 and 2). To quantify local pair interactions between individuals during movements, we calculated two variables for each horse pair, namely (i) pairwise distance $d$, and (ii) movement similarity $C$, i.e., directional correlation between trajectories of horse pairs[36,37]. Both variables were assessed over each 5-min observation session and averaged over the five 5-min observation sessions (see Methods).

From its founding in 1997, all individuals of the Hortobágy population have been individually identified and monitored by the reserve staff. Monitoring includes recording individuals' life-histories (e.g., births, deaths), parentage confirmed by genetic sampling, associations of individuals to sub-units of the multilevel social system (i.e., harems) and changes in harem memberships[14]. These population monitoring data have typically spanned for timescales of years to decades but are sparse and less detailed (updated monthly), in contrast to the detailed and fine-scale but short-term drone observations (Fig. 1d; see Methods). We investigated how these two data with different timescales relate to each other, and whether the latter can provide a robust measurement for the multilevel social structure.

Due to the long-term population monitoring, group dynamics in the population, including temporal development of harems, dispersal of individuals between harems, and associations of individuals in the same harem was known, not only at the time of movement tracking, but also for the previous 21 years prior to the drone observations as well as for the subsequent 2 years (Fig. 2). Some harems existed for more than a decade, and their composition typically changed slowly over time (Fig. 2). Individuals may have spent several years in the same harem, during which time they may have developed familiarity with each other, which may influence their behaviour also after leaving their harem and already belonging to different harems. To quantify familiarity, we used $t_{past}$, the time a pair of individuals has been together in the same harem in the 2 years prior to the movement observations (see Methods). From the genetic sampling the population's genealogy could be reconstructed, and thus the kinship between each pair of individuals was determined (Fig. 3).

### Levels of the society

We first assessed the relationship between motion patterns and the multilevel social structure of the herd (associations of individuals to sub-units was known from long-term population monitoring). Social levels, as expected[2], were associated with different levels of cohesion between their members: pairwise distance was lower ($p < 0.0001$) and movement similarity was higher within harems than among harems within the whole herd ($p < 0.0001$, $n_1 = 711$, $n_2 = 21374$, randomisation tests, see Methods for details; see Supplementary Table 2; Fig. 4b, c, Supplementary Fig. 1). Since juveniles stay in the parental harem for several years[27,28], we considered an adult female and its subadult, not yet dispersed offspring as a sub-unit within harems and called it "family" (note that foals still dependent on their mothers were not included in "families"). We found that pairwise distance within "families" was typically lower ($p < 0.001$) and movement similarity was higher than in harems ($p < 0.0001$, $n_1 = 70$, $n_2 = 711$, randomisation tests, Fig. 4b, c, Supplementary Fig. 1). Although this sub-unit in equids (i.e., a female and its juvenile offspring) is usually not considered as a separate social level[2], its high cohesion shows similarity to a social level, nested within harems. We investigated the behaviour of bachelor males as well, but in this case only pairs of a bachelor male and a harem member individual could be considered, because bachelor males could not be individually identified and thus it was not possible to

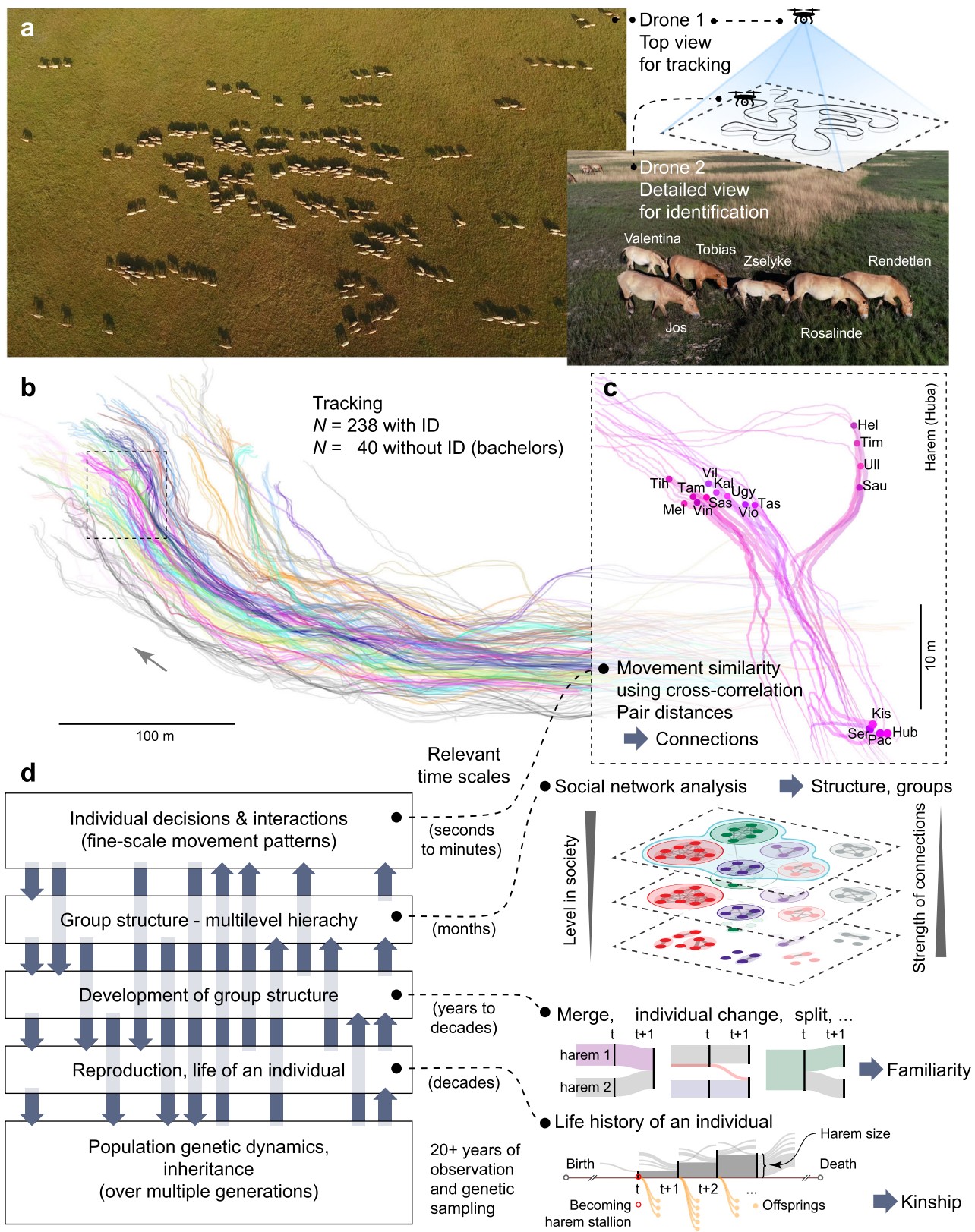

distinguish intra-group and inter-group bachelor pairs. We found lower cohesion between a bachelor group and a harem group than between two harem groups, as pairwise distances were higher ($p < 0.0001$) and movement similarities were lower ($p < 0.0001$, $n_1 = 8778$, $n_2 = 20959$, randomisation tests, Fig. 4b, c) between a bachelor and harem member individual than between two harem-living individuals belonging to different harems.

Considering the features of collective movement, we found that the herd's multilevel structure was obvious in the $d-C$ plots, both for a single 5-min and for averaged sessions, as individual pairs belonging to the same harem were separated from the pairs belonging to different harems, primarily along the axis $d$ (Supplementary Fig. 2a, b). Accordingly, the distribution of pairwise distances clearly showed two peaks, where the first peak corresponded to the harem level, while the

**Fig. 1 | Overview of the main concept and the data acquisition technique.**
**a** Sample images and a sketch of the setup for 4k filming of Przewalski's horses at Pentezug reserve, Hortobágy National Park, Hungary, in 2018 with two drones. The higher drone provides a large-scale top view for tracking individuals and the background to get coordinates and movement in an earth-fixed coordinate system (**b**). The lower drone scans the area with horses to get a detailed view for individual recognition. **b** Example trajectories of all horses belonging to the population (*n* = 278) from a 5-min long drone recording. Arrow shows the main direction of motion of the herd. Individuals (known identification, *n* = 238) are colour-coded

based on the group they belong to (out of 31 harems), or shown as grey in the case of bachelor males (i.e., males that are not part of a harem, *n* = 40). **c** Detailed view of trajectories of a single harem, with all individuals shown with dots at a given point in time. Three letter codes show their identities. **d** Diagram of the main concept (on the left) showing important aspects of collectively living animals, and the complex interplays between these components. The components may have a relevant temporal scale (shown in the middle) that spans through several orders of magnitudes (from seconds and minutes to several decades). A detailed schematic explanation of each component is provided on the right.

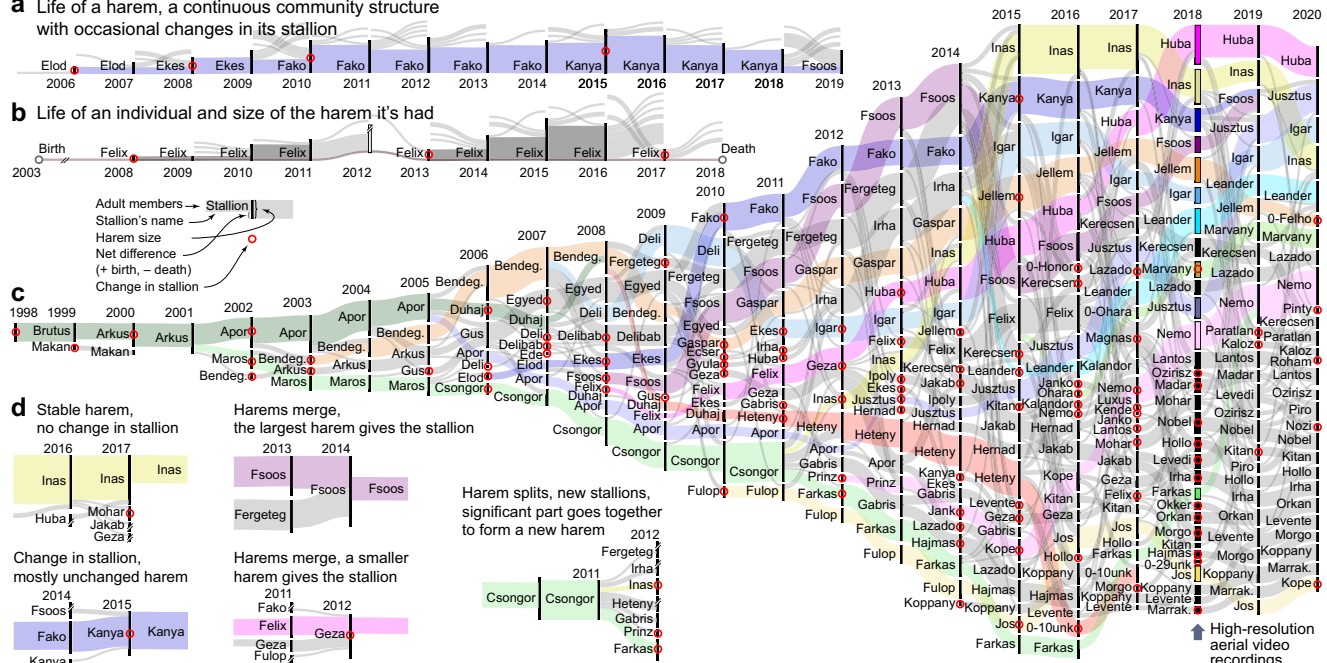

**Fig. 2 | Group dynamics and the temporal development of harem structure of Przewalski's horses at Hortobágy National Park (Hungary) between 1997 and 2020. a, b** Sample timeline from the point of view of a harem (**a**) and of an individual harem stallion (**b**). For each year, the length of the vertical heavy black line indicates the total number of members, with in-coming (and out-going) edges pointing from (and towards) other harems. All adult members are presented except

bachelors. Names indicate the harem stallion, red circles indicate stallion changes. **c** Data visualisation from the long-term population monitoring from 1998 to 2020. For 2018, when the high-resolution aerial observations were recorded, colour indicates several harems with matching colour in Fig. 4d. The longest lasting harems are indicated by coloured lines. **d** Examples of important events related to changes in harem structure or to emergence of a new harem stallion.

second peak to the herd level (Fig. 4a). Although the two peaks were more obvious for averaged sessions, the two peaks were noticeable even in a single 5-min session (Supplementary Fig. 2c–e). Therefore, pairwise distances during collective movements averaged over several minutes are enough to detect multilevel structure of horse herds, and to classify individuals into sub-units based solely on their movement without any prior knowledge on their social relationships (Supplementary Fig. 3, see Supplementary Note 1 for details).

## Society, kinship and familiarity
As pair distances during collective movements were clearly related to the social structure, we built proximity networks of harems and individuals to characterise the herd's multilevel society. These networks were based on averaged distances between individuals during movements, i.e., network edges represent typical distances less than a given threshold (see Methods; Fig. 4d, Supplementary Fig. 4). We investigated how this social network relates to kinship (Fig. 3) and familiarity of individuals (i.e., duration of common past membership for a pair of individuals, $t_{past}$; Fig. 2).

First, we studied the bonds between adult females within harems. Here, kinship did not seem to influence the structure of the social network, as we found no significant differences in network distances if we compared close kin, i.e., full or half-sibling ($p = 0.079$, $n_1 = 15$,

$n_2 = 199$) and parent-offspring, adult female pairs to more distant relatives ($p = 0.334$, $n_1 = 9$, $n_2 = 199$, randomisation tests; see Supplementary Note 2 for additional info). Furthermore, the harem choice of females seemed not to be affected by kinship, as sibling female pairs were not found more frequently in the same harem than in the whole population ($p = 0.107$, $n = 31$, randomisation test). Familiarity had, however, an effect on network distances within harems, as adult female pairs were closer to each other in the network if they spent more time in the same harem in the previous 2 years (Pearson's $r = -0.175$, $p = 0.006$, $n = 199$, using randomisation; excluding close kin).

Next, we studied the bonds between the harems. The network distance between harems was shaped by stallion kinship, as harems of sibling stallions were located closer in the network than harems of more distantly related stallions ($p < 0.001$, $n_1 = 53$, $n_2 = 411$, randomisation test, Fig. 5a, b; note that shorter network distances mean shorter spatial distances and usually also higher movement similarities). The harems of full sibling stallions were even closer than the half-siblings' harems ($p = 0.046$, $n_1 = 9$, $n_2 = 44$, randomisation test, Fig. 5a, b). Sibling relations of stallions could not be separated from familiarity, as group membership of stallions while being bachelors was not known. However, common past membership of stallions in the parental harem at young age did not seem to affect proximity of their harems, as

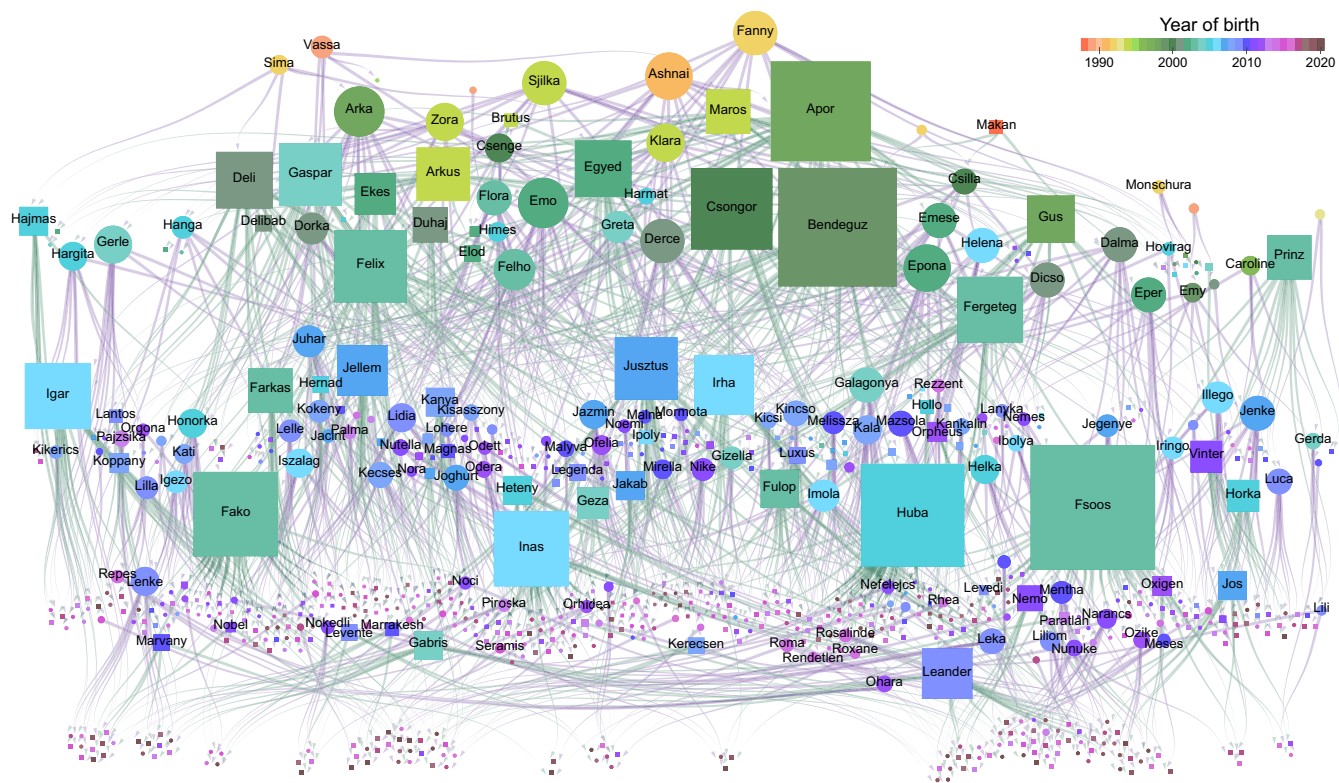

**Fig. 3 | Genealogy of the Przewalski's horse population at Hortobágy National Park between 1997 and 2020.** Nodes correspond to individuals living in the reserve (circles for females and squares for males) from the establishment of the population in 1997 until the end of 2020. The node size is proportional to the total number of their offspring, node colour indicates the year of birth. Edges point from parent to offspring (purple for mother, green for father). Due to the parentage database kinship relation between each pair of horses is known.

harems of non-related stallions were not closer in the network if the stallions lived longer in the same parental harem in the past (Pearson's $r = −0.035$, $p = 0.260$, $n = 381$, randomisation test). Network distances of harems were associated with female kinship too, as harems containing full or half-sibling adult female pairs were usually located closer to each other than the harems, which contain only more distantly related females ($p = 0.004$, $n_1 = 65$, $n_2 = 206$, randomisation test; excluding harems with familiar females). The network distance between harems with parent-offspring adult female pairs did not, however, differ significantly from harems not containing close kin females ($p = 0.342$, $n_1 = 14$, $n_2 = 206$, randomisation test; excluding harems with familiar females).

## Group dynamics

We investigated whether the network among harems is related to group dynamics. First, we asked whether the females' harem changes can be predicted on the basis of this network, i.e., changes happen more frequently between neighbouring harems or not. The network distance of harems that had exchanged females in the 2 years prior to the movement observations was lower than the distance between other harems ($p = 0.021$, $n_1 = 27$, $n_2 = 438$), and the same was true for harems that exchanged females in 2 years following the observations (i.e., in the "future", $p = 0.017$, $n_1 = 24$, $n_2 = 441$, randomisation tests, Fig. 5c, d). Next, we focused on the individuals and asked whether we can predict which females are going to change harem. We quantified the time a pair of females spent in the same harem in the subsequent 2 years following the movement observations ($t_{future}$). When investigating the future of females that were currently harem-mates, we found that if they were closer to each other in the network then they typically spent more time in the same harem in the subsequent 2 years following the observed movements (Pearson's $r = −0.160$, $p = 0.012$, $n = 199$,

randomisation test; excluding close kin). When investigating the future of females from different harems, we assessed movement similarities, since pairwise distances are primarily determined by the location of their harems. Interestingly, the adult female pairs from different harems, which later became harem-mates (for at least 3 months) in the subsequent 2 years following the movement observations, already had more similar movement paths than their female harem-mates' average ($p = 0.007$, $n = 109$, for non-related females with $t_{past} = 0$, $t_{future} > 90$ days vs. average movement similarity of females from their harems with $t_{future} = 0$, randomisation test).

## Network centrality

To reveal further details of the harem network, and hence of the possible origins of herd formation, we studied how different harem traits were related to network centrality—a network metric characterising importance of nodes (Supplementary Fig. 5). We found that a harem's closeness centrality (i.e., the reciprocal of the mean shortest path distance from all other reachable nodes) was positively associated with the harem's age (Pearson's $r = 0.600$, $p < 0.001$, $n = 31$) and the harem stallion's experience in harem keeping (i.e., the number of years the stallion has had a harem, in total; Pearson's $r = 0.663$, $p < 0.0001$, $n = 31$, randomisation tests; Fig. 5e, Supplementary Fig. 6a, c). These latter two variables were also related to each other, suggesting that older harems typically belong to more experienced stallions (Pearson's $r = 0.724$, $p < 0.0001$, $n = 31$, randomisation test; Supplementary Fig. 7a; see Supplementary Note 3 for additional info). Unsurprisingly, we found a connection between the harem's closeness centrality and the stallion's age as well (Pearson's $r = 0.631$, $p < 0.001$, $n = 30$; Supplementary Fig. 6e), because a stallion's harem keeping experience is in strong correlation with its age (Pearson's $r = 0.849$, $p < 0.0001$, $n = 30$, randomisation tests). The size of a harem including adult and subadult

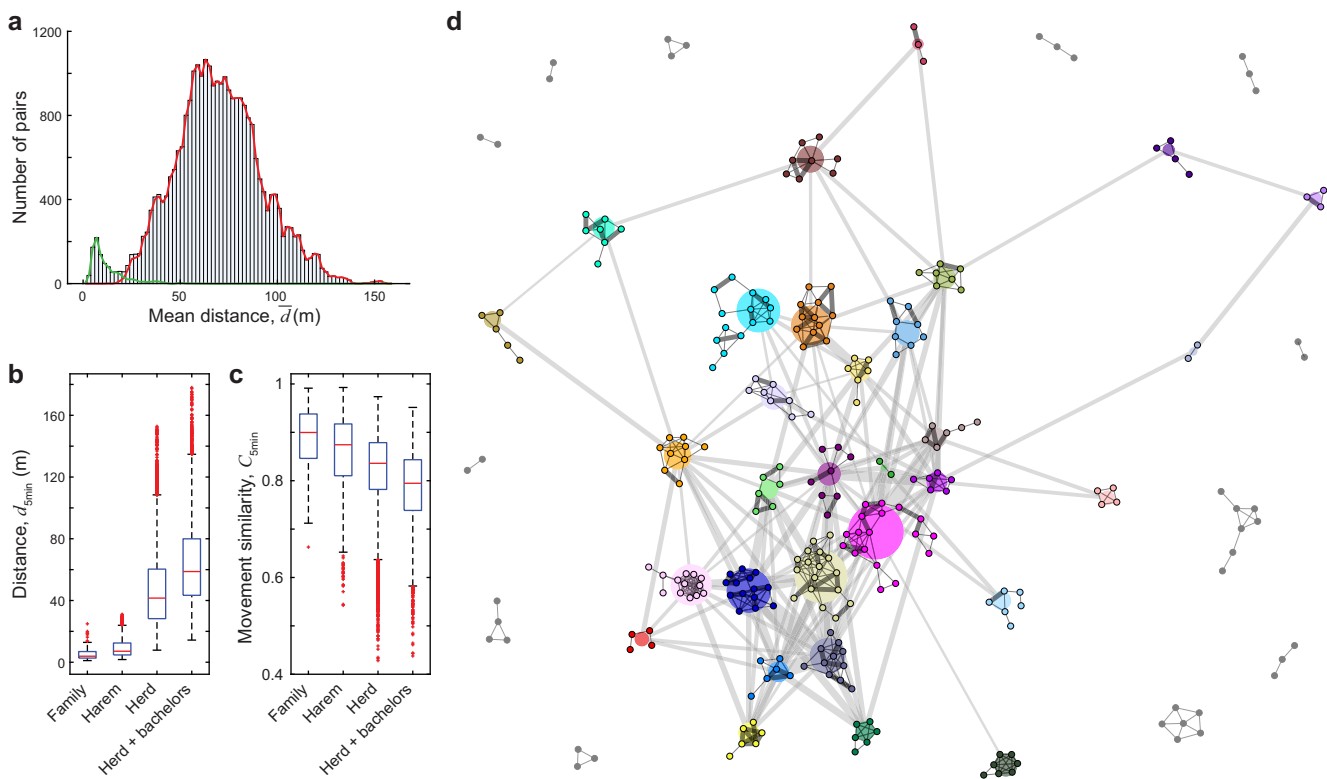

**Fig. 4 | Social network of the Przewalski's horse herd based on local pair interactions during collective movements. a** Histogram of pair distances between horses during collective movements averaged over five 5-min observation sessions (bars), for intra-harem (green line) and inter-harem pairs of individuals (red line); the two peaks correspond to the two social levels of the multilevel society: harem and herd. **b, c** Pairwise distance and movement similarity of horse pairs belonging to the (i) same family (i.e., a female and its 1–5 years old pre-dispersal offspring, $n = 70$ horse pairs), (ii) different family but same harem ($n = 707$), (iii) different harem (same herd, $n = 20959$); and (iv) of a bachelor and a harem-living individual pair ($n = 8778$), in a single 5-min observation session. The pairs' distance increases while movement similarity decreases from family to herd including bachelors. Boxes range from the 25th to 75th percentile, while central

marks denote medians, whiskers extend to extreme data points not considered outliers (marked in red). **d** Multilevel social network of the herd, presented as two networks overlaid, based on average proximity during observed collective movements. *Top layer:* Nodes (small circles) correspond to individuals, while colour denotes the harem they belong to or shown with grey for bachelors. Bonds (black lines) are based on averaged pairwise distances between individuals: an edge is drawn between two individuals if their averaged distance $\bar{d} < d_{th}$ (for further details, see main text). Heavy connections depict close family ties. *Bottom layer:* Social network of harems, nodes (larger circles) correspond to harems (with matching colours of the individuals and size proportional to harem size), and edges (grey lines) show spatial proximity between harems.

members was also related to the harem's closeness centrality (Pearson's $r = 0.468$, $p = 0.005$, $n = 31$; Supplementary Fig. 6g), although this was not the case if counting only the adult members (Pearson's $r = 0.286$, $p = 0.061$, $n = 31$, randomisation tests). On the other hand, we found that average distance of the closest bachelor male to harems was not related to the closeness centrality (Pearson's $r = -0.133$, $p = 0.241$, $n = 31$), although the harems with more adult members (i.e., containing more adult females) were typically further from the bachelors (Pearson's $r = 0.430$, $p = 0.010$, $n = 31$, randomisation tests; Fig. 5f).

## Discussion

In this study, we simultaneously tracked all individuals of a large herd of Przewalski's horses and showed that analysing their collective movements for a few minutes is sufficient to determine the current harem membership of the individuals and to infer past and future social dynamics of the population.

Despite our analyses being based on a few minutes long recordings, fine-scale tracking and high-throughput data driven analysis of movements uncovered novel social relationships among the horses. Our results on intra-harem bonds primarily provided by familiarity are consistent with known sociality of polygynous equids[38]. Inter-harem bonds, and thus the formation of large herds,

however, is not fully understood in equids[12–14]. Therefore, more research is needed to determine the possible preconditions and factors that may lead to massive herds from independent harems. Our results suggest that aggregation of harems is associated with kinship in this Przewalski's horse population, and male-male and female-female sibling relations are both important. Similarly, in plains zebras (*Equus quagga*), a closely related species, aggregation of harems is driven by kinship, but only by female-female kin relations[39]. In other systems, such as human and other primate societies, association of groups can be based on male-male bonds, which may or may not be associated with genetic relatedness[4,40,41]. We expected association between kinship and aggregation of harems, since both the relatedness among harem stallions and adult females may strengthen inter-harem tolerance and thus reinforce herd formation[42]. Note, however, that stallion kinship could not be separated from familiarity in this study, because there is a possibility that genetically related males were formerly group-mates while being bachelors, and developed familiarity with each other. Female familiarity probably also contributes to harem aggregation, due to the member transfers between harems[42]. On the other hand, female transfers were more frequent between nearby harems, which suggests a reinforcing effect between female exchange and proximity of harems.

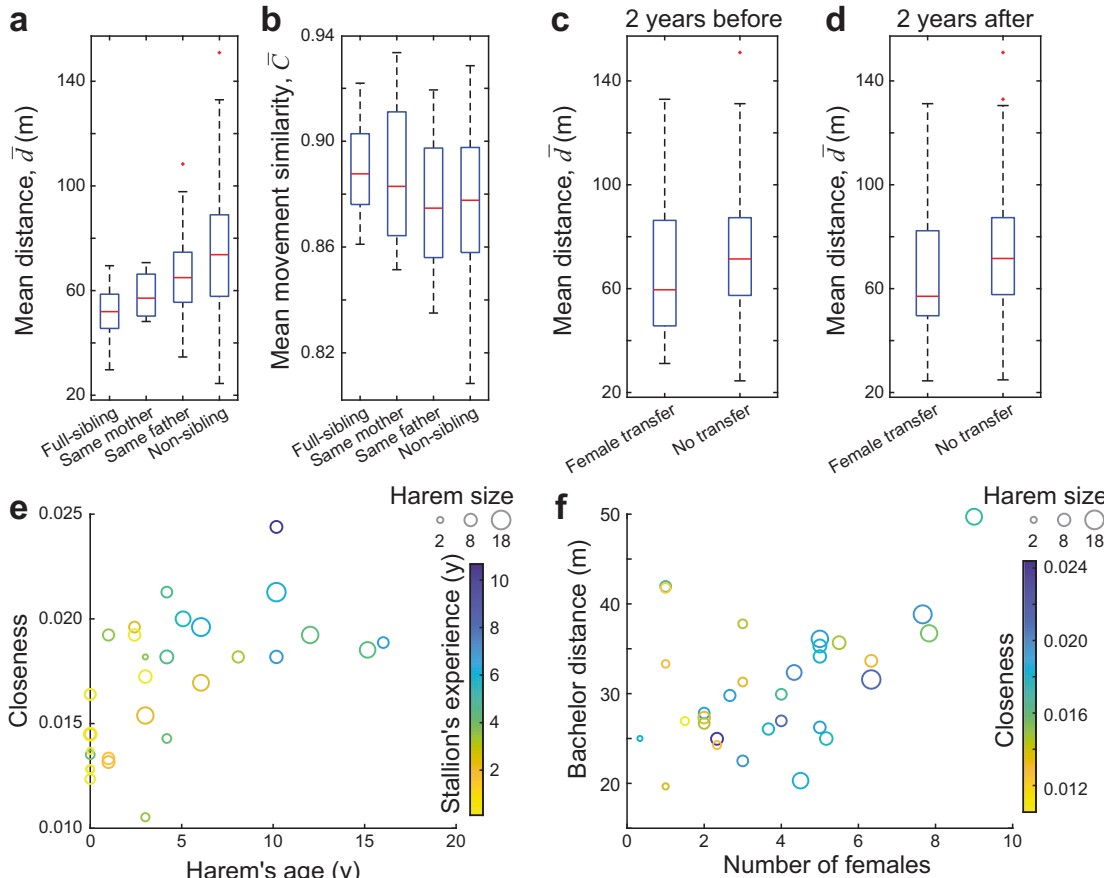

**Fig. 5 | Social and group dynamic processes in connection with the social network. a, b** Distance and movement similarity of harems, averaged over five 5-min observation sessions, for different kinship relationships between their stallions: for full sibling ($n = 9$ harem pairs), maternal ($n = 4$) and paternal half-sibling ($n = 40$), and more distantly related stallions ($n = 411$). **c, d** Harems between which female transfer occurred in the 2 years prior (**c**) or following (**d**) the movement observations were closer (and had more similar movement) than harems without female transfer ($n_{prior,transfer} = 27$, $n_{prior,no\ transfer} = 438$, $n_{following,transfer} = 24$,

$n_{following,no\ transfer} = 441$ harem pairs). Boxes range from the 25th to 75th percentile, while central marks denote medians, whiskers extend to extreme data points not considered outliers (marked in red). **e** Closeness centrality of harems in the network against the harem's age in years. Node size denotes the harem size, while node colour the stallion's experience of harem keeping (i.e., how long the stallion kept a harem in years). **f** Average distance of the closest bachelor male to harems against the number of adult female members in the harem. Node size denotes the harem size, while node colour the closeness centrality of harems.

Relationship between network centrality and harem traits, namely that older and larger harems with older and more experienced stallions occupy more central locations in the network, may suggest benefits of central positions. Network centrality implies vicinity of more harems and their stallions, which, due to communal defence, may promote protection from bachelor males. In this way, stallions of central harems may reduce the risk of takeovers and thus prolong their tenure, while females may reduce male harassment and the risk of infanticide[43,44], which were both observed in the studied population (V.K., pers. obs.). Moreover, centrality might influence exposure to horseflies and other biting insects. On the other hand, centrality of older harems may be an emergent property as well, since newly formed harems are often temporarily isolated and join the herd later[14]. In some primates multilevel group formation is similarly driven by conspecific social threats, e.g., bachelor threat and infanticide[42,45,46]. Also, herd size of plains zebras is shown to depend on environmental and social factors, such as the number of bachelor males[12]. Accordingly, this suggests that herd formation in Hortobágy might be the result of an interplay between several factors, such as kinship, familiarity and coalitions of harem stallions against bachelor threat, and may also be driven by females, however, this suggestion needs further investigation.

Familiarity (i.e., past associations in the same harem) affecting movements of individuals was expected, as social bonds develop over time. On the other hand, the relation between current movement and

future associations, namely that based on movement it can be predicted which female pairs will become harem-mates later or which ones will stay in the same harem, years after the observations, is surprising.

Our findings can be useful in population monitoring, e.g., to determine the number and size of harems in a herd, which would probably work for other Przewalski's horse or feral horse populations as well. Applicability in other species, however, strongly depends on the dynamics of the animals' movement and group stability (e.g., in fission-fusion societies longer observations and/or more occasions would be necessary to detect the multilevel structure). Although we only studied herd movements here, tracking individuals while foraging or resting would likely lead to interesting insights into what kind of social interactions hold individuals together or space them apart, e.g., how exactly the affiliative or agonistic interactions influence the spatial organisation in a herd.

Drones provide a precise and low-disturbance method for behavioural and movement tracking of animals in the wild, however their application has been rare to date[47–51], possibly due to technological difficulties of simultaneous tracking. Automatic tracking methods, like machine learning, would offer new perspectives in the study of complex social systems and their collective behaviour[51]. Our study highlights how relatively short observations of movements can reveal detailed social dynamics of a population not just in the present, but in

the past and future, and moreover may provide a unique tool to expand our knowledge on development and function of complex societies.

## Methods

This study was approved by the Government Office for Hajdú-Bihar County (Hungary) under the reference number HB-03/KTF/00779-24/2017, and by the Hortobágy National Park Directorate under the reference number 3482-2/2017.

### Study area and population

We studied Przewalski's horses in Hortobágy National Park, located in eastern Hungary (47°31'3.3"N 21°5'34.1"E). In 1997, Przewalski's horses were introduced to the Pentezug reserve, a 3000 ha steppe area in Hortobágy National Park, as part of an ecological habitat management scheme by large grazers: wild horses and cattle. The area is surrounded by an electric fence that prevents migration of large grazers but does not limit the native fauna. The conditions are close to wild, human activities are restricted, and the horses are not fed or watered. The total number of large grazers was around 680 during our observation period (273–278 Przewalski's horses and cca. 400 Heck cattle). The Przewalski's horse population originated from 31 founder individuals which arrived in the reserve between 1997 and 2017 from different European zoological gardens, though during the observations only two founder individuals were still alive. The formation of social groups and mate choice are natural, without any human intervention. Since the population is closed, birth control was used from 2013 treating females with the immunocontraception vaccine porcine zona pellucida[52]. The inbreeding coefficient in the population is relatively high (0.176 during the study year)[14], due to a strong bottleneck effect caused by the extinction in the wild of this species in the 1960s[53].

From the beginning, horses have lived in year-round stable harems, consisting of a single male, several adult females and their young. Female offspring leave the natal harem at the age of 2–3 years and join other harems; males leave at the age of 2–4 years and join with other non-breeding but already dispersed males and former harem stallions to form single-sex bachelor groups. In the first years, harems and bachelor groups stayed isolated, and used non-overlapping home ranges; then later, and also at the time of our observations, all harem and bachelor groups united into a massive herd with a common home range[14].

### Population monitoring

The Przewalski's horse population in the Pentezug reserve has been monitored regularly since its founding in 1997. All individuals except young males while bachelors are individually identified on the basis of natural pelage colouring and characteristic features and injuries, supported by a photo catalogue, harem composition lists and DNA-fingerprint database[14]. Data collection includes records on life history events, like birth date, death date, identity of parents and changes in harem membership. Biopsy samples are taken from each individual at the age of 1 year, which serves for determining parentage and updating the DNA-fingerprint database. Parentage records based on observations are supported by DNA-tests in the majority of cases (74%). Males are individually known until leaving the parental harem, then again when they acquire a harem or when they die, but membership information is lacking while in bachelor groups, which have a less consistent membership. Harem composition lists are updated with a temporal resolution of 12 +/− 8 (mean +/− SD) observations per year. Individual traits known from population monitoring include age, sex, role in the social system (adult female, subadult individual in parental harem, harem stallion, bachelor), relatedness to other individuals, and monthly updated records on harem membership.

### Drone observations

We performed aerial video recordings of the herd on the move with two DJI Phantom 4 drones simultaneously at 4k resolution and 25 fps (frames-per-second). Prior to the study we tested the disturbance of drones with decreasing flying altitudes and observed that horses started to avoid drones at around 3–4 m flying altitudes. A drone flying high (100–300 m from the ground) recorded the moving herd and provided the image for movement tracking, while a drone flying low (10–30 m from the ground) scanned through the whole herd and provided a detailed image for individual identification (Fig. 1a). The recordings of the two drones were synchronised, and thus identities of individuals could be matched with the tracked trajectories. The top-view drone's video was processed with the Motion Tracking function of Blender v2.79b[54] and the global position of each individual was determined in every second frame, i.e., with 12.5 fps temporal resolution. By tracking several fixed points of the background, we solved the camera motion and reconstructed the tracking scene, then projected the horse movement tracks to the 3D view of the background. We set a ground level based on three background-fixed points, a local origin and $x$-axis, and scaled the background with two distinctive landscape items (e.g., a well, a solitary tree, etc.) by measuring the distance between them on Google Maps. This way we obtained horse coordinates in metres in a background-fixed coordinate system. The $x$–$y$ coordinates of a horse were defined on its withers (the ridge between the shoulder blades) at $z = 1.25$ m altitude (the average height of Przewalski's horses). Coordinates of horses in each frame were exported from Blender to 'csv' files with a custom-written script in Python 2.7. The accuracy of horse positions in the background-fixed system was +/− 0.2 m (the average noise when tracking a stationary point). Identities of individuals were determined on the lower drone's videos by one observer based on natural colouring and characteristic marks.

We recorded 320 s of continuous movements (called as observation sessions) of the herd when most of the individuals were moving continuously through the whole session between feeding and drinking places or towards a dust bath (300–600 m travel distance). The study contains five observation sessions, recorded on five different days, 1–2 weeks apart (August 17, August 24, September 5, September 13, and October 2 in 2018), during daylight hours (between 8 a.m. and 3 p.m.). Average travelling velocity was around 0.85 m/s, the horses were moving at a pace from a walk to a gallop. The population size on the first observation day was 278 individuals, in total, of which 238 were individuals belonging to harems and 40 were bachelor males. We aimed to track the whole population and to identify all harem-living individuals, but in some cases a few individuals were not possible to recognise or did not appear on the video (Supplementary Table 1). As the bachelor males could not be identified individually, we denoted their tracks as "unidentified bachelor". The size of the population slightly decreased during our aerial observations due to a few deaths, and was 273 on the last observation day (Supplementary Table 1). Identification reliability, i.e., the ratio of individuals that got the same ID during repeated identification attempts, was 93%.

### Data analyses

**Movement variables.** We calculated pairwise movement variables, pairwise distance ($d$) and movement similarity ($C$) over two timescales: (i) a "5-min" (i.e., 320 s) observation session, where a high-resolution drone video recording was used for tracking at 12.5 fps (resulting in 4000 "frames", timestamped locations for each individual visible during a session), (ii) averaged sessions, i.e., the averaged data over five 5-min observation sessions recorded on different days. We defined the 5-min distance ($d_{5min}$) between two individuals as the distance of the pair in metres averaged over a 5-min recording session. Averaged distance $\bar{d}$ is the average over the five 5-min sessions. We calculated movement similarity ($C_{5min}$) between a pair of individuals as the

directional correlation with a time delay between their trajectories[36,55], for a 5-min session. Trajectories were smoothed using Gaussian smoothing ($\sigma = 1.2$ s). For every pair the entire duration (320 s; 4000 frames) of the trail was used (as a single time window) and the highest correlation was chosen using all possible time delays in the range of [−16 s, 16 s] with a step of 1 frame (0.08 s). Pairwise correlation values were omitted, where the correlation was negative. For all other aspects, the steps of this movement analysis were identical to previous studies[56]. Robustness against the values of the parameters were analysed in the previous studies, but we rechecked it here as well. Averaged movement similarity $\bar{C}$ was obtained as the average over the five sessions. Bachelor males were considered only in the analyses of 5-min sessions and excluded from the analyses of averaged sessions, since their identity could not be matched through different observation days.

**Society.** The single breeding male in a harem is called the "harem stallion", aged between 6 and 15 years ($n = 31$). Females that have already dispersed from their natal harem or are at least 2 years old and their harem stallion is not their father due to a stallion change, are considered "adult females" ($n = 115$). A "bachelor" is a non-breeding male, including young males, which have dispersed from their natal harem and have not gained a harem yet, and old males, which previously had a harem ($n = 40$). A family is defined as an adult female and her pre-dispersal offspring except foals born in the year of observation, where the offspring may be 1–5 years old, and the father of the offspring may be one or multiple stallions. A harem consists of a harem stallion, several adult usually non-related females associating with the stallion, and their pre-dispersal offspring. Individuals in the study were assigned to a harem based on the group composition records at the first observation day. Harem size denotes the number of all individuals belonging to the harem, both adult and subadult, while adult harem size counts the adult females and the harem stallion (there were no multi-male harems). In the pairwise analyses bachelor-bachelor pairs were excluded, as bachelors could not be identified individually, and hence we could not classify them as same-group or different-group pairs.

**Social network.** To study the multilevel social structure, we constructed a proximity network of harems and proximity networks of individuals within each harem, based on pairwise distances of individuals. In the network of harems an edge connected two harems if the averaged distance between any of their members was less than a threshold ($\bar{d} < d_{th}$; Supplementary Fig. 4). In the analyses we used the harem network with the smallest $d_{th}$ that ensures a single connected component ($d_{th} = 53$ m; Fig. 4d). Also within each harem, we defined a network of individuals in a similar manner, where two individuals were connected with an edge if their averaged distance was smaller than a threshold, $\bar{d} < d_{th,harem}$. Since typical distances varied among harems, $d_{th,harem}$ was determined separately for each harem as the smallest threshold, where all members of the harem were connected to the network (Fig. 4d). When determining $d_{th,harem}$ we excluded females that changed harem during the observation period, and hence, were members of multiple harems on different observation days. Distance in the social network of individuals and harems was calculated as the length of the shortest path between the nodes.

In our study, we chose an arbitrary $d_{th}$ distance threshold when calculating the social network. To test the robustness of the results on the $d_{th} = 53$ m harem network (association between network centrality and different harem traits), we also investigated networks with 10% higher and lower $d_{th}$ and obtained similar results (see Supplementary Note 4 for details).

**Kinship and familiarity.** For the association of the social network with kinship, we tested whether close relatives (i.e., offspring-parent, full

and half-sibling pairs) behave differently than individuals, which are more distantly related. We defined the following kin relationships: full siblings (both parents are common), half-siblings with same mother, half-siblings with same father, parent-offspring (mother and daughter), and more distantly related (i.e., all other individuals). For quantifying familiarity, we used the group composition lists between 1997 and 2020 to calculate shared membership of horse pairs (i.e., both individuals present in the same harem). For adult females, familiarity ($t_{past}$) was defined as the number of days in the same harem by a pair in the 2 years prior to the movement observations (between 2016 and 2018). Similarly, $t_{future}$ was the number of days a female pair spent in the same harem in the 2 years following the movement observations (between 2018 and 2020). Note that for $t_{past}$ and $t_{future}$ shared time in the same harem can refer to multiple harems (i.e., association with several different stallions), in this case the number of days in the same harem is summed up for all common harems. For harem stallions, $t_{past*}$ was the number of days the stallion pair spent in the same harem during young age while being subadults. Note that since bachelor group compositions were not recorded during population monitoring, time spent in the same bachelor group could not be quantified for stallions.

When investigating the effect of kinship and familiarity, we tried to separate these effects, as far as possible, by analysing certain subsets of the data. Harem stallions and adult females were analysed separately, and adult females were further divided into females from the same harem, and females from different harems. The effect of kinship was studied in subsets where individual pairs did not share past membership in the previous 2 years ($t_{past} = 0$), except in case of stallions; and the effect of familiarity was studied in subsets where closely related individual pairs (parent-offspring, full sibling, half-sibling) were excluded.

We considered female kinship between two harems in the following harem pair subsets: (i) harem pairs containing full and/or half-sibling inter-harem adult female pairs but not containing parent-offspring inter-harem adult female pairs, (ii) harem pairs containing parent-offspring inter-harem adult female pairs but not containing full or half-sibling inter-harem adult female pairs, and (iii) harem pairs containing only more distantly related inter-harem adult female pairs. To quantify the familiarity of adult females between two harems, we calculated $t_{past\_af\_harem}$, as the sum of $t_{past}$ for all possible pairs of adult females from the two harems. When studying the association between female kinship and harem distances, we compared two of the above subsets; to exclude the effect of familiarity, only harem pairs were considered where $t_{past\_af\_harem} = 0$.

**Group dynamics.** We considered only adult females changing their breeding harems, i.e., excluded harem changes of young females from their natal harem to the first breeding harem, in the 2 years prior (2016–2018) and after (2018–2020) the movement observations. By comparing the membership data of 2 years (e.g., memberships in 2016 and 2018), we counted the number of female transfer between each harem pair, regardless of direction (i.e., if one female moved from harem A to B and one female from B to A that means two changes happened between harems A and B).

**Harem traits.** We approached harems as dynamically changing "communities" and considered them as the same "community" unless a given amount of change in the membership occurred. When studying the development of harems, we used one observation per year from the group composition records of Hortobágy National Park, the one prior to 1st September in each year. In 1997 the first record defined the initial harems. Then, in each subsequent year $i$, we calculated the Jaccard index between the adult members (2-year-old and older females and the stallion) of all harems in year $i−1$ and year $i$. The ancestor of a harem in year $i$ is the harem in year $i−1$ with the maximal Jaccard index,

but with at least two common adult individuals. If a harem in year $i-1$ has more descendant harems in year $i$, the one with the most common individuals will be the descendant, or the oldest one if the common part is equal in size. If a harem in year $i$ does not have an ancestor in year $i-1$, then it is assigned as a new harem with a starting date of the first observation; if a harem in year $i-1$ does not have a descendant in year $i$, then the harem is assigned as ended. Harem ages were calculated from the harems' starting dates.

A male's harem keeping experience is defined as the time in years during which the male kept the position of being a harem stallion. If a stallion had more than one harem throughout its life, the harem keeping time is summed up for all harems.

**Statistical tests.** Randomisation tests were performed as follows: (i) a test statistics is calculated, i.e., (a) a mean difference between two samples of a variable, (b) a mean difference between paired samples of a variable, (c) a correlation coefficient between two variables; (ii) the empirical dataset is rearranged randomly, i.e., (a) the two samples are pooled then divided again randomly to two samples according to the original sample sizes, (b) and (c) paired data points in one of the variables are permuted randomly, while the other variable is unchanged; then the test statistics is calculated on the permutated data; this point is repeated (number of iterations was $n = 10,000$ in all tests) to estimate the test statistics' probability distribution; (iii) $p$-value is obtained from the cumulative distribution function of the test statistics (i.e., at the empirical value of test statistics). All randomisation tests were one-sided.

Calculations and statistical analyses were performed in MATLAB R2021a[57] and CUDA 11.5. For data organisation and storing MATLAB R2021a and Microsoft Excel for Mac 16.54 were used.

### Reporting summary
Further information on research design is available in the Nature Portfolio Reporting Summary linked to this article.

## Data availability
Data generated during the analyses that support the findings of this study have been deposited on Github at https://github.com/katalinozogany/wildhorse_mls. The raw data are available under restricted access for nature conservation reasons, access can be obtained from the Hortobágy National Park Directorate and the first author on reasonable request.

## Code availability
Custom codes used in the analyses have been deposited on Github at https://github.com/katalinozogany/wildhorse_mls.

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

## Acknowledgements

We thank the Hortobágy National Park Directorate—in particular directors Zita Kovács and Gergely Á. Medgyesi—for supporting the research. We are grateful to Tímea Szabados for assisting in field observations and Miklós Bán for database organisation. We are grateful to Sarah R. B. King, Dóra Bíró, Daniel I. Rubenstein and Tamás Vicsek for commenting and revising the manuscript. We thank Tamás Vicsek, Gábor Vásárhelyi and Gergő Somorjai for initial ideas concerning the field observations. We thank all persons assisting in the field and students participating in video processing. This research and K.O. was supported by the National Research, Devel-opment and Innovation Fund of Hungary financed under the FK 123880 funding scheme. A.F. was supported by the Romanian Ministry of Research, Innovation and Digitisation (CNCS-UEFISCDI, PN-III-P1-1.1-TE-2021-0502) and by a Postdoctoral Advanced Fellowship from the Babeș-Bolyai University (CNFIS-FDI-2022-0179). Z.B. was supported by the Thematic Excellence Programme (TKP2021-NKTA-32). M.N. acknowledges support from the Hungarian Academy of Sciences, Grant 95152 (to the MTA-ELTE "Lendület" Collective Behaviour Research Group) and the MTA-ELTE Statistical and Biological Research Group and Eötvös Loránd University.

## Author contributions

K.O., M.N. and Z.B. designed the main concepts of the study; K.O. designed and performed the movement observations and tracking; V.K. collected the population monitoring data; K.O. and V.K. curated the data; M.N. designed and performed the trajectory analyses; K.O. and M.N. designed and performed the data analyses and the visualisations; all authors contributed to the interpretation of the results; K.O. wrote the initial manuscript, A.F., Z.B. and M.N. edited the final manuscript.

## Funding

## Competing interests

The authors declare no competing interests.
