## [Peer Review File · Nature Communications]

REVIEWER COMMENTS

Reviewer #1 (Remarks to the Author):

This manuscript reports on an exciting study where authors simultaneously track the position and movement of many wild horses, revealing how relatively little (and simple) data/metrics can provide robust quantification of their social relationships (pair relationships and harems' structure). What is more, these simple measures can also predict future changes in social dynamics. I think the work undertaken is excellent. The figures and images are outstanding, the methods and analyses appropriate, and the use of supplemental figures/details to provide more detailed information is commendable. However, I find the narrative and writing poor; I do not think the write-up of the work does the hard work or study findings justice.

The problem with the narrative is most problematic in the introduction to the study. The background and rationale lack context and detail. In fact, after reading the introduction I found difficult to understand/follow what the authors were studying and why. In their revised MS, authors need to explain to the reader what they expected at the outset and the approach they took - i.e. explain Figure 1D... For example, did they think that the harems would be so easily identified from simple distances among horses? Or that the movements of individuals would predict future associations (i.e., Figure 2)? If so, why? If they didn't expect this, and found this out when exploring the data, then tell this narrative instead.

Related to the above point, I think the authors focus on multi-level societies in the introduction should be reduced/removed. There are lots of systems that have multi-level societies. The important contribution of the work/approach is in the combination of high-resolution movement data with longer term information. The authors show that with only minutes of data, they can reliably predict not only the number and size of harems within this population, but also say something informative about the age of stallions/harems for those identified. To generate this information "on the ground" would require many hours of focal and scan observations, and lots of people. Therefore, from a population monitoring and management tool, this is outstanding. This approach must be highlighted and introduced at the outset (and in the title).

I therefore recommend the authors spend more time in the introduction talking about how researchers traditionally quantify social interactions and social systems, and if and how "snapshots" (see recent Royal Society meeting - <https://royalsociety.org/science-events-and-lectures/2022/05/animal-behaviour/> and Biro et al. 2016 TREE) of the whole system may afford the same information, quicker and reliably. Opinion/review articles on the topic a few years ago e.g., King et al. (2018 TREE) suggested "technological advances, providing researchers with the opportunity to quantify and model the heterogeneity that exists within the social groupings they study and within the environments in which

these groups live” but don’t think these authors/others thought such “little” amount of high-resolution data could be so informative (even more so when paired with more “traditional” methods for context) as shown here.

In contrast to the introduction, the results and discussion does a much better job of providing context and importance of the work and approach. I provide some more detailed comments, mostly related to the first half of the manuscript, below.

Line 1. The title of the paper is quite descriptive and vague. Given that the authors high throughput data driven analysis reveals past group dynamics and predict future ones; I would argue this is the novel contribution that should be highlighted, if possible (see main points, above).

Line 26-27. Multilevel societies, are common across different animal systems, and not sure “intricate” is the correct description here? What is intricate about it? Also, should so much text be devoted to ML society (see main comments, above).

Line 29. comma or dash needs to be inserted before and after “the last wild horses”

Line 37-38. “Movement variables also predicted exchange of members in the future.” Can this sentence be restructured so that the reader knows how movement variables predicted exchange? What do changes to they predict, how?

Line 44. Which types of societies are more or less complex in nature? What makes ML among the most complex? See main comments.

Line 49. Change “but it is” to “but are”

Line 50. Remove “even in” before birds.

Line 51. If ML societies are so widespread across taxa, is it “unique”? I suggest removing.

Line 59. Explain defence polygyny as a mating system, briefly.

Line 66. “and” aggregate not “or”.

Line 58-72. I think this paragraph introducing the system and specific population should come after the next paragraph which attempts to layout the questions addressed.

Line 72. “to explore the fine structure and dynamics of a multilevel society”. This is very vague statement – better used to start the next paragraph?

Line 73-82. There are no citations to back up statements in this paragraph and need to be added. This paragraph is also incredibly vague. What is central power? What is control? The spatial associations among individuals and their collective movements are a cause and consequence of the social structure so I am not sure what authors are proposing here. The aims need to be much clearer and precise. Can the authors set out some more concrete information on what they wanted to investigate and why? Same for the next paragraph - Line 83-89: What are “the attributes of collective movement”? what are

the “lesser-known details about the horses ML society”? What are movement interactions? What are future social dynamics?

Line 83. What is “on a fine scale”? I think some detail on the spatial and temporal resolution of data is needed here, and maybe introduce Figure 1 at this point. Line 91-106 gives details on the different types of data, but I think again, this does not provide enough detail or context for the reader. I would expect this paragraph to be a narrative for the information provided in Figure 1 which is really good. Authors need to highlight that they have intense and detailed data at a fine-scale over short time periods, and coarse and less detailed data over long term. The authors plan to see how the former relates to the latter, and whether and if the former can provide a robust measurement of the ML dynamics in the system.

Line 118. This expectation has not been laid out.

General: how well do authors think their approach might work for understanding the structure and dynamics of social relationships in more stable social groups?

Reviewer #2 (Remarks to the Author):

Key results:

Unravelling mechanisms and causes leading to multi-level societies, all species confounded including our own, are essential to understand sociality. Publications on the subject are exceedingly rare. Among these few, even fewer studies have high-precision data such as Ozogany’s et al.’s that result in the ability to show how the relationship between respective individuals’ spatial positions and collective movements, relate to kinship and previous familiarity as well as network centrality in a large herd.

Validity:

There are no flaws that could prevent the manuscript's publication according to my evaluation.

Originality

As stated above, the study's originality and significance lies in its contribution to the knowledge of mechanisms and possible causes leading to a multi-level society, and therefore sociality in general, through high-precision data. The results span multiple scientific disciplines, notably in social sciences.

Data and methodology

This field study conducted on a free-living large herbivore, the Przewalski's horse, does not pretend to an experimental approach. The methodology and data contains unprecedented detailed information on ca. 250 individuals living in one herd. The methodological approach is certainly reproducible, either on a different or the same population, knowing that individuals in the same population would have aged. Method description and data access would allow any researcher to reproduce analyses or re-analyze the data.

Statistics

- Error-bars are present in all figures where adequate, and explained in the methods section including in Supplementary Information.
- All statistical tests are appropriate. Personally, I prefer to use non-parametric tests on data with a non-normal distribution instead of Gaussian transformations necessary to use parametric tests such as Pearson's correlation, but then, Gaussian transformation procedure, through normalizing data distribution, is standard practice.
- Having been involved in a PhD concerning social networks a few years ago, I realized that social network analysis is a fast moving field, and I presently feel out of date. I suggest the authors contact Damien Farine or one of his research associates to validate up to date procedures.

Conclusions

Conclusions are adequate with regard to results, but see below.

Suggested improvements:

General comments: The manuscript including the abstract could benefit from being more concise and better structured around the main research questions.

Suggested title: Kinship and familiarity relate to a multi-level society in Przewalski's horses.

Introduction: Main research questions could be reformulated at the end of the introduction.

Methodology: no remarks but see below

Results: The result part contains elements of methodology and should be readjusted.

Discussion: The discussion could benefit from a broader framework for future research:

The high precision spatial data was collected during collective herd movements. What about the relevance for data collected during resting or foraging ?

Of course it would be important to study social interactions between individuals to explain mechanisms leading to spatial distances and movement coordination. What kind of social interactions hold individuals together or space them out, eg. what affiliative or agonistic interactions influence the spatial organization in a herd?

I have checked all items on the review checklist and replied to questions where relevant.

Due to the importance of understanding sociality in multi-level society, including our species, and the exceedingly rare high-precision data of this study, I recommend a revised version of the manuscript to be published in Nature Communications.

Publication in 'Nature Ecology and Evolution' or 'Scientific Reports' could be a second option, but would miss out on a larger public impact, especially in the social sciences.

We thank the Reviewers for their highly constructive comments. We generally agree with the points that they have made, and we have modified the manuscript in accordance with their recommendations. We have completely rewritten the Introduction.

There follows a detailed list of responses (written in blue italic font) to each Reviewer's evaluation, together with an account of the changes we have made to the text (highlighted in red).

REVIEWER COMMENTS

Reviewer #1 (Remarks to the Author):

This manuscript reports on an exciting study where authors simultaneously track the position and movement of many wild horses, revealing how relatively little (and simple) data/metrics can provide robust quantification of their social relationships (pair relationships and harems' structure). What is more, these simple measures can also predict future changes in social dynamics. I think the work undertaken is excellent. The figures and images are outstanding, the methods and analyses appropriate, and the use of supplemental figures/details to provide more detailed information is commendable.

A1. We would like to thank the Reviewer for the appreciation of our work. We thank also generally for the detailed suggestions and constructive criticism through their review. We provide answers to each comment and suggestion point-by-point, below.

However, I find the narrative and writing poor; I do not think the write-up of the work does the hard work or study findings justice.

The problem with the narrative is most problematic in the introduction to the study. The background and rationale lack context and detail. In fact, after reading the introduction I found difficult to understand/follow what the authors were studying and why. In their revised MS, authors need to explain to the reader what they expected at the outset and the approach they took - i.e. explain Figure 1D...

A2. We have worked intensively to improve the narrative, especially in the Introduction, which we completely rewrote and added new details to the context of the study with the appropriate references (see also answers A5 and A6). The last paragraph of the revised Introduction (line 78-89) summarizes our study approach, the questions we raised at the outset and the main results that we found during assessing the data. Later, in the detailed description of our approach entitled "Data acquisition techniques" in the revised MS, we include additional explanation of different data used in the study (line 114-119).

For example, did they think that the harems would be so easily identified from simple distances among horses?

A3. We include additional information at line 82-84.

Or that the movements of individuals would predict future associations (i.e., Figure 2)? If so, why? If they didn't expect this, and found this out when exploring the data, then tell this narrative instead.

A4. *Thanks for the comment. We added that the prediction of future changes is an unexpected finding at line 88.*

Related to the above point, I think the authors focus on multi-level societies in the introduction should be reduced/removed. There are lots of systems that have multi-level societies. The important contribution of the work/approach is in the combination of high-resolution movement data with longer term information. The authors show that with only minutes of data, they can reliably predict not only the number and size of harems within this population, but also say something informative about the age of stallions/harems for those identified. To generate this information “on the ground” would require many hours of focal and scan observations, and lots of people. Therefore, from a population monitoring and management tool, this is outstanding. This approach must be highlighted and introduced at the outset (and in the title).

A5. *We thank the Reviewer for highlighting in which aspect they found our work to be most impactful. We agree that our work provides important new applications for population monitoring, and we revised the title as well. However, we would like to point out that we think our work provides important contribution to the studies of multilevel societies (MLS), which view is shared by Reviewer #2, as expressed in their review as a primary contribution (see text before answer A23). So, when rewriting the revised version, we aimed to find a balance between the suggestions expressed by the two Reviewers. As a compromised solution, we decided to focus less on MLS – and in parallel put more emphasis on the importance of providing new technologies – while, for example, writing the first sentence of the Abstract (line 29-30) and in the first sentence of the Introduction (line 45-46). Furthermore, the first paragraph of the Introduction – which gives background to MLS - is shortened (line 46-54). We highlighted our approach of combining high resolution data with long term monitoring in line 116-118. We also highlighted that movement analysis can predict past and future social dynamics in the title of the MS.*

I therefore recommend the authors spend more time in the introduction talking about how researchers traditionally quantify social interactions and social systems, and if and how “snapshots” (see recent Royal Society meeting - <https://royalsociety.org/science-events-and-lectures/2022/05/animal-behaviour/> and Biro et al. 2016 TREE) of the whole system may afford the same information, quicker and reliably. Opinion/review articles on the topic a few years ago e.g., King et al. (2018 TREE) suggested “technological advances, providing researchers with the opportunity to quantify and model the heterogeneity that exists within the social groupings they study and within the environments in which these groups live” but don’t think these authors/others thought such “little” amount of high-resolution data could be so informative (even more so when paired with more “traditional” methods for context) as shown here.

A6. *Thank you for the explicit suggestions. We agree and we added a new paragraph to the Introduction concerning this recommendation (line 68-77), including the mentioned references as well.*

In contrast to the introduction, the results and discussion does a much better job of providing context and importance of the work and approach. I provide some more detailed comments, mostly related to the first half of the manuscript, below.

Line 1. The title of the paper is quite descriptive and vague. Given that the authors high throughput data driven analysis reveals past group dynamics and predict future ones; I would argue this is the novel contribution that should be highlighted, if possible (see main points, above).

A7. *We modified the title of the MS (see also answer A5) – while aiming to find a reasonable compromise with the Reviewer #2 recommendations, who suggested the title “Kinship and familiarity relate to a multi-level society in Przewalski’s horses” – and now we highlight the predicting power of our research related to past and future social dynamics in the revised title.*

Line 26-27. Multilevel societies, are common across different animal systems, and not sure “intricate” is the correct description here? What is intricate about it? Also, should so much text be devoted to ML society (see main comments, above).

A8. *We removed the word “intricate” and substantially reduced the description of MLS in the Abstract (see also answer A5), e.g., the first two sentences of the original Abstract, mentioned in this comment, were deleted.*

Line 29. comma or dash needs to be inserted before and after “the last wild horses”

A9. *The phrase “the last wild horses” was removed, while shortening the Abstract to meet the formal requirements of Nature Communications.*

Line 37-38. “Movement variables also predicted exchange of members in the future.” Can this sentence be restructured so that the reader knows how movement variables predicted exchange? What do changes to they predict, how?

A10. *We rephrased this sentence into “High movement similarity of females from different harems predicted becoming harem mates in the future.”*

Line 44. Which types of societies are more or less complex in nature? What makes ML among the most complex? See main comments.

A11. *This is a valid and rather philosophical question. Although many publications refer to multilevel societies as “complex” or “most complex” societies (see e.g. [1,2,6,15,21]). In the revised version, we reduced the text on MLS and the number of mentioning the complexity or uniqueness of these societies.*

Line 49. Change “but it is” to “but are”

A12. *Done.*

Line 50. Remove “even in” before birds.

A13. *Done.*

Line 51. If ML societies are so widespread across taxa, is it “unique”? I suggest removing.

A14. *Done.*

Line 59. Explain defence polygyny as a mating system, briefly.

A15. *We added a short explanation (line 56-59).*

Line 66. “and” aggregate not “or”.

A16. *We rephrased the sentence to make it clearer (line 60-63).*

Line 58-72. I think this paragraph introducing the system and specific population should come after the next paragraph which attempts to layout the questions addressed.

A17. *The next paragraph of the original MS, mentioned here, was removed as suggested below (answer A19).*

Line 72. “to explore the fine structure and dynamics of a multilevel society”. This is very vague statement – better used to start the next paragraph?

A18. *This sentence was deleted.*

Line 73-82. There are no citations to back up statements in this paragraph and need to be added. This paragraph is also incredibly vague. What is central power? What is control? The spatial associations among individuals and their collective movements are a cause and consequence of the social structure so I am not sure what authors are proposing here. The aims need to be much clearer and precise. Can the authors set out some more concrete information on what they wanted to investigate and why? Same for the next paragraph - Line 83-89: What are “the attributes of collective movement”? what are the “lesser-known details about the horses ML society”? What are movement interactions? What are future social dynamics?

A19. *These two paragraphs of the original MS were removed from the revised version. Instead, we added a new paragraph with a brief summary of our approach and the studied questions (line 78-89).*

Line 83. What is “on a fine scale”? I think some detail on the spatial and temporal resolution of data is needed here, and maybe introduce Figure 1 at this point. Line 91-106 gives details on the different types of data, but I think again, this does not provide enough detail or context for the reader. I would expect this paragraph to be a narrative for the information provided in Figure 1 which is really good. Authors need to highlight that they have intense and detailed data at a fine-scale over short time periods, and coarse and less detailed data over long term. The authors plan to see how the former relates to the latter, and whether and if the former can provide a robust measurement of the ML dynamics in the system.

A20. *We added more information for the spatial and temporal resolution of movement data in line 100. We expanded the paragraph of the original MS about the different types of data and it became a separate subsection entitled “Data acquisition techniques”. We introduce Figure 1 in this subsection, and we highlight how we combined data with different timescales in line 113-118.*

Line 118. This expectation has not been laid out.

A21. *A reference is added where this expectation is found in detail.*

General: how well do authors think their approach might work for understanding the structure and dynamics of social relationships in more stable social groups?

A22. In most cases, individuals of the same core unit are spatially closer to each other than to individuals from other core units in multilevel societies, thus core units are spatially delineated and may be determined from pair distances in other systems as well. Also studying path similarity and quantifying temporal delay between pairs of individuals by using directional correlation were shown to be a powerful tool for several different animal groups (pigeons, fish, humans, etc.). We think the approach presented here for Przewalski's horses would work well for other animal collectives. If the groups are stable, we expect that it would work robustly. Although the exact dynamics of movements may be systems specific, thus for some animal groups longer observations or more occasions would be needed to apply a similar approach. We included a paragraph at lines 294-301.

Reviewer #2 (Remarks to the Author):

Key results:

Unravelling mechanisms and causes leading to multi-level societies, all species confounded including our own, are essential to understand sociality. Publications on the subject are exceedingly rare. Among these few, even fewer studies have high-precision data such as Ozogany's et al.'s that result in the ability to show how the relationship between respective individuals' spatial positions and collective movements, relate to kinship and previous familiarity as well as network centrality in a large herd.

Validity:

There are no flaws that could prevent the manuscript's publication according to my evaluation.

Originality

As stated above, the study's originality and significance lies in its contribution to the knowledge of mechanisms and possible causes leading to a multi-level society, and therefore sociality in general, through high-precision data. The results span multiple scientific disciplines, notably in social sciences.

A23. We would like to thank the Reviewer for their high evaluation of our manuscript. We are very grateful for all their comments and detailed recommendations, which we answer separately below.

Data and methodology

This field study conducted on a free-living large herbivore, the Przewalski's horse, does not pretend to an experimental approach. The methodology and data contains unprecedented detailed information on ca. 250 individuals living in one herd. The methodological approach is certainly reproducible, either on a different or the same population, knowing that individuals in the same population would have aged. Method description and data access would allow any researcher to reproduce analyses or re-analyze the data.

A24. We confirm that we will publish the high-resolution trajectory data and our analysis codes without any restriction when our manuscript is published.

Statistics

- Error-bars are present in all figures where adequate, and explained in the methods section including in Supplementary Information.
- All statistical tests are appropriate. Personally, I prefer to use non-parametric tests on data with a non-normal distribution instead of Gaussian transformations necessary to use parametric tests such as

Pearson's correlation, but then, Gaussian transformation procedure, through normalizing data distribution, is standard practice.

A25. Thank you for the detailed comments. In the revised version we use only randomisation tests when reporting p values of the statistical tests.

- Having been involved in a PhD concerning social networks a few years ago, I realized that social network analysis is a fast moving field, and I presently feel out of date. I suggest the authors contact Damien Farine or one of his research associates to validate up to date procedures.

A26. We thank the Reviewer for suggesting Damien Farine. We think highly of his and his collaborators' pioneering works in quantitative understanding of social systems, including multilevel societies in animal group. However, two of us have a background in statistical physics and studying complex systems, so we feel confident in the network analysis performed. We have decided not to contact Dr. Farine, as that may have significantly extended the time needed to prepare the revised version.

Conclusions

Conclusions are adequate with regard to results, but see below.

Suggested improvements:

General comments: The manuscript including the abstract could benefit from being more concise and better structured around the main research questions.

A27. We revised considerably the MS, especially the Introduction which we completely rewrote, to be more clear and better structured. Please, find the details above, in our responses to Reviewer #1.

Suggested title: Kinship and familiarity relate to a multi-level society in Przewalski's horses.

A28. We modified the title of the paper, however, we needed to find a reasonable compromise with Reviewer #1 who suggested to remove the focus on multilevel societies and emphasize instead the high throughput data driven analysis. On the other hand, we found the phrase "familiarity" so appropriate for this variable that in the revised MS we started to use it instead of the previous "shared past" or "shared history".

Introduction: Main research questions could be reformulated at the end of the introduction.

A29. In the last paragraph of the revised Introduction (line 78-89) we summarize our approach, the questions raised at the outset, and the main results found during assessing the data.

Methodology: no remarks but see below

Results: The result part contains elements of methodology and should be readjusted.

A30. A new subsection entitled "Data acquisition techniques" was added in the revised MS, which summarizes the main methodology essential for understanding the Results. Some methodology details were moved to this "Data acquisition techniques" subsection in the revised version, e.g. line 104-109, which were earlier in the Results section.

Discussion: The discussion could benefit from a broader framework for future research:

The high precision spatial data was collected during collective herd movements. What about the relevance for data collected during resting or foraging?

Of course, it would be important to study social interactions between individuals to explain mechanisms leading to spatial distances and movement coordination. What kind of social interactions hold individuals together or space them out, eg. what affiliative or agonistic interactions influence the spatial organization in a herd?

A31. *We added a text incorporating these remarks to the Discussion (line 293-296).*

I have checked all items on the review checklist and replied to questions where relevant.

Due to the importance of understanding sociality in multi-level society, including our species, and the exceedingly rare high-precision data of this study, I recommend a revised version of the manuscript to be published in Nature Communications.

Publication in 'Nature Ecology and Evolution' or 'Scientific Reports' could be a second option, but would miss out on a larger public impact, especially in the social sciences.

A32. *Many thanks again for your very positive comments.*

REVIEWERS' COMMENTS

Reviewer #1 (Remarks to the Author):

This version is excellent. The authors have responded to all reviewer comments and made sensible changes where appropriate; I enjoyed reading the revision and responses.

Authors should give a final check for clarity of the text throughout before providing the final version to the journal.

Best wishes

Reviewer #2 (Remarks to the Author):

Remarks on the second version of the manuscript by Ozogany et al.

The Authors greatly improved the manuscript with regard to its first version and integrated both reviewer's comments, compromising where adequate.

Due to the importance of understanding sociality in multi-level society, including our species, and the exceedingly rare high-precision data of this study, I therefore recommend the manuscript to be published in Nature Communications.

The following comments are partly a repetition of my remarks on the first draft.

Key results:

Unravelling mechanisms and causes leading to multi-level societies, all species confounded including our own, are essential to understand sociality. Publications on the subject are rare. Even fewer studies have high-precision data such as Ozogany's et al.'s that result in the ability to show how the relationship between respective individuals' spatial positions and collective movements, relate to kinship and previous familiarity as well as network centrality in a large herd.

Validity:

There are no flaws that could prevent the manuscript's publication according to my evaluation.

Originality

As stated above, the study's originality and significance lies in its contribution to the knowledge of mechanisms and possible causes leading to a multi-level society, and therefore sociality in general, through extremely rare high-precision data. The results span multiple scientific disciplines, notably in social sciences.

Data and methodology

This field study conducted on a free-living large herbivore, the Przewalski's horse, does not pretend to an experimental approach. The methodology and data contains unprecedented detailed information on ca. 250 individuals living in one herd. The methodological approach is certainly reproducible.

Statistics

- Error-bars are present in all figures where adequate, and explained in the methods section including in Supplementary Information.

- All statistical tests are appropriate.

.

Conclusions

Conclusions are adequate with regard to results.

I have checked all items on the review checklist and replied to questions where relevant.

Detailed comments on the text:

Line 49: cross out 'polygynous', in some species they are not

Line 49: In mammals, they are best known....

Line 51: cross out 'birds', as the article does not discuss them

Line 120: replace 'migration' by 'dispersal'?, or simply 'moving' ?

Line 146: replace "because of lacking the identity"?

Line 272 -274: The benefit of a central position in the herd could have other causes than a reduced risk of bachelor male harassment, such as lower exposure to horse-flies (tabanids) or other biting insects.

Line 275: Young foals suffering from sometimes mortal wounds inflicted by conspecifics do not necessarily die from intentional infanticide, a prerequisite for its definition in behavioural ecology. Some very young foals are accidentally wounded when they get in the middle of a fight between adults. So the sentence could be modified, unless the author was a direct witness of events.

We would like to thank very much the Reviewers for their encouraging comments and all the useful suggestions that helped to improve the manuscript. We agree with their points, and we have modified the manuscript accordingly.

There follows a detailed list of responses (written in blue italic font) to each Reviewer's evaluation, together with an account of the changes we have made to the text (highlighted in red in the MS file).

REVIEWER COMMENTS

Reviewer #1 (Remarks to the Author):

This version is excellent. The authors have responded to all reviewer comments and made sensible changes where appropriate; I enjoyed reading the revision and responses.

Authors should give a final check for clarity of the text throughout before providing the final version to the journal.

Best wishes

A1. We would like to thank the Reviewer again for highlighting the parts where the manuscript needed to be improved and all their detailed recommendations that helped significantly to clarify the narrative. Before submitting the final version, we checked the manuscript and corrected where needed.

Reviewer #2 (Remarks to the Author):

Remarks on the second version of the manuscript by Ozogany et al.

The Authors greatly improved the manuscript with regard to its first version and integrated both reviewer's comments, compromising where adequate.

Due to the importance of understanding sociality in multi-level society, including our species, and the exceedingly rare high-precision data of this study, I therefore recommend the manuscript to be published in Nature Communications.

The following comments are partly a repetition of my remarks on the first draft.

Key results:

Unravelling mechanisms and causes leading to multi-level societies, all species confounded including our own, are essential to understand sociality. Publications on the subject are rare. Even fewer studies have high-precision data such as Ozogany's et al.'s that result in the ability to show how the relationship between respective individuals' spatial positions and collective movements, relate to kinship and previous familiarity as well as network centrality in a large herd.

Validity:

There are no flaws that could prevent the manuscript's publication according to my evaluation.

Originality

As stated above, the study's originality and significance lies in its contribution to the knowledge of mechanisms and possible causes leading to a multi-level society, and therefore sociality in general, through extremely rare high-precision data. The results span multiple scientific disciplines, notably in social sciences.

Data and methodology

This field study conducted on a free-living large herbivore, the Przewalski's horse, does not pretend to an experimental approach. The methodology and data contains unprecedented detailed information on ca. 250 individuals living in one herd. The methodological approach is certainly reproducible.

Statistics

- Error-bars are present in all figures where adequate, and explained in the methods section including in Supplementary Information.
- All statistical tests are appropriate.

Conclusions

Conclusions are adequate with regard to results.

I have checked all items on the review checklist and replied to questions where relevant.

A2. *We thank very much the Reviewer for appreciating our work and for the motivating evaluation in their review. We thank also for all the helpful corrections and ideas suggested in their review that greatly contributed to clarify and improve the manuscript. We provide answers to each suggestion point-by-point, below.*

Detailed comments on the text:

Line 49: cross out 'polygynous', in some species they are not

A3. *We removed the word 'polygynous' from the sentence.*

Line 49: In mammals, they are best known....

Line 51: cross out 'birds', as the article does not discuss them

A4. *We thank the Reviewer for these suggestions, but we feel it is important to point out in the Introduction that multilevel societies may occur also in other taxa, and not only in mammals.*

Line 120: replace 'migration' by 'dispersal'?, or simply 'moving' ?

A5. *We agree that 'dispersal' is a better term and replaced 'migration' by 'dispersal', accordingly. We thank for the suggestion.*

Line 146: replace ‘because of lacking the identity’?

A6. *We rephrased this part of the sentence to ‘because bachelor males could not be individually identified’.*

Line 272 -274: The benefit of a central position in the herd could have other causes than a reduced risk of bachelor male harassment, such as lower exposure to horse-flies (tabanids) or other biting insects.

A7. *We completely agree that centrality in the herd can have other benefits, than just the reduced risk of male harassment. Thus, we added a sentence about the effects of biting insects in line 273-274.*

Line 275: Young foals suffering from sometimes mortal wounds inflicted by conspecifics do not necessarily die from intentional infanticide, a prerequisite for its definition in behavioural ecology. Some very young foals are accidentally wounded when they get in the middle of a fight between adults. So the sentence could be modified, unless the author was a direct witness of events.

A8. *We agree that this is an intriguing issue. An author of this paper (V.K) was a direct witness of intentional infanticides, some of them were documented with photos and videos, and in some cases also the identity of the attacking individual could be determined. Although, these observations are not published yet. Of course, sometimes foals may die from accidental wounds, when getting in the middle of a fight between adult horses, which were also recorded in the studied population. But the wounds of the two cases are clearly different (e.g. in the case of infanticide a bite on the neck close to the mane is typical, while accidental wounds are primarily caused by kicks). In the manuscript we moved upward and rephrased a half sentence considering the observations of infanticides in our population ‘which were both observed in the studied population (V.K., pers. obs.)’ to the line 272-273.*